# The Knowledge of Malnutrition—Geriatric (KoM-G) 2.0 Questionnaire for Health Care Institutions: Cross-Cultural Adaptation into German, Czech, Dutch and Turkish

**DOI:** 10.3390/nu16091374

**Published:** 2024-04-30

**Authors:** Silvia Bauer, Jan Pospichal, Viviënne Huppertz, Vit Blanar, Bulent Saka, Doris Eglseer

**Affiliations:** 1Department of Nursing Science, Medical University of Graz, Neue Stiftingtalstraße 6, 8010 Graz, Austria; 2Faculty of Health Studies, University of Pardubice, Studentska 95, 532 10 Pardubice, Czech Republic; 3Department of Respiratory Medicine, School of Nutrition and Translational Research in Metabolism (NUTRIM), Maastricht University, Universiteitssingel 50, 6202 AZ Maastricht, The Netherlands; 4Department Internal Medicine, Istanbul Faculty of Medicine, Istanbul University, Millet Str., Çapa, Fatih, 34093 Istanbul, Turkey

**Keywords:** cross-cultural adaptation, knowledge, malnutrition, questionnaire, nursing staff, geriatric

## Abstract

It is necessary for nursing staff to have adequate knowledge of malnutrition in older people in order to provide high quality care. This study was conducted to update the Knowledge of Malnutrition—Geriatric (KoM-G) questionnaire to fit different settings and to cross-culturally adapt it to the German, Czech, Dutch and Turkish languages. In Part 1 of the study, the KoM-G questionnaire was updated and adapted for use in different settings. Content validation of the KoM-G 2.0 was carried out in a Delphi study with 16 experts. The final KoM-G 2.0 questionnaire consists of 16 items with a Scale Content Validity Index/Average of 94.5%. In Part 2, the English KoM-G 2.0 was cross-culturally adapted into the German, Czech, Dutch and Turkish languages. In the pilot test, between 96.9% (The Netherlands) and 97.8% (Austria) of the nursing staff rated the items as understandable. The KoM-G 2.0 is an up-to-date questionnaire with a highly satisfactory Content Validity Index. It was cross-culturally adapted into the German, Czech, Dutch, and Turkish languages, and the understandability was high. At the moment, the necessary comprehensive psychometric testing of the KoM-G 2.0 is in process. Afterwards it can be used to compare nurses’ knowledge between various countries and settings.

## 1. Introduction

Malnutrition in older persons is recognized as a highly prevalent problem. A recent meta-analysis indicated pooled prevalence rates of malnutrition risk in European countries of 28.0% in hospitals, 17.5% in nursing homes and 8.5% in community settings [1]. Prevalence rates also varied widely between countries. For example, Spain had a prevalence rate of 15.2% in hospitals and Turkey, of 27.7%, which may partly be explained by their use of different screening tools [1].

If a patient is malnourished, several consequences can be expected, including longer hospital stays, complications such as prolonged wound healing, increased morbidity and mortality, as well as high health care costs [2,3]. Malnutrition often affects older persons due to their age-related changes in appetite and the senses of smell and taste, as well as their difficulties in chewing and swallowing. In addition, older people with malnutrition tend to have a low socio-economic status, which is also associated with nutritional status [4,5]. Several international guidelines have been published that describe the adequate management of malnutrition in older persons [6,7,8], but the adherence to such guidelines is often limited in clinical practice [9,10].

Most experts believe that health care staff must have adequate knowledge to be able to properly interpret and implement guideline recommendations [11,12,13]. Put another way, a lack of knowledge with regard to malnutrition is frequently mentioned as a barrier that prevents health care staff from conducting evidence-based interventions with persons at risk of malnutrition [14,15,16]. However, recently published studies show that about 30% of bachelor’s degree programs offered at European nursing education institutions and about 50% of European medical universities do not even include the topic of malnutrition in their curricula. These studies identified a lack of malnutrition education in several topics, especially in multidisciplinary cooperation, different nutritional interventions and the evaluation of these interventions [17,18]. The fact that the topic of malnutrition is rarely addressed in nurse and physician education may lead to poor knowledge and awareness and may subsequently, besides other factors like resources, management, etc., add to inadequate nutritional care.

Questionnaires can be used effectively to examine knowledge of a specific topic in particular populations [19]. As the numbers of multi-national and multi-cultural studies increase, it becomes increasingly important to provide adequate translations and cross-cultural adaptations of these questionnaires. By adequately and cross-culturally adapting instruments, identical research can be carried out in different cultures. In addition, such adaptation is more cost-effective than the development of new instruments [20]. It is, however, necessary to properly cross-culturally adapt an existing instrument and to analyze the psychometric properties of the instrument in the target language before applying the instrument [21].

The knowledge of Malnutrition-Geriatric (KoM-G) questionnaire was systematically developed and psychometrically tested in 2012 for use among nursing staff in nursing homes in Austria [22]. An update of this questionnaire was recommended, because new guidelines on nutritional care for older people with malnutrition have been published since 2012 (e.g., by the European Society for Clinical Nutrition and Metabolism (ESPEN)). These guidelines should be included in state-of-the-art knowledge assessment instruments [6,8]. Furthermore, an adaptation to other health care settings was needed and a cross-cultural adaptation of the questionnaire was warranted in order to collect and compare data from various settings and countries.

To our knowledge, only a few questionnaires to assess malnutrition knowledge among nursing staff are available. These few available questionnaires in use interchanged the roles of members of different professions, combined measurements of knowledge and attitudes and were neither systematically developed nor comprehensively psychometrically tested [11,23,24].

As a result, an update and cross-cultural adaptation of the KoM-G to the KoM-G 2.0 aims to provide health care professionals, researchers and educators with a systematically developed tool in different languages. This tool can then be psychometrically tested and, if psychometric properties are proven to be adequate, can be used to identify areas of adequate knowledge as well as knowledge gaps.

Therefore, this study was carried out to update the KoM-G questionnaire and to cross-culturally adapt it to the German, Czech, Dutch and Turkish languages. The study was performed in a project consisting of two parts:

Part 1: Updating the KoM-G questionnaire, adapting it for use in different settings and validating the content of the resulting English KoM-G 2.0.

Part 2: Cross-culturally adapting the KoM-G 2.0 to the German, Czech, Dutch and Turkish languages.

## 2. Materials, Methods and Results

### 2.1. Design

This study used a psychometric methodological design [25] and consisted of two parts.

#### 2.1.1. Part 1: Updating the KoM-G Questionnaire, Adapting It for Use in Different Settings and Validating the Content of the Resulting English KoM-G 2.0

In this first step, the original English KoM-G was updated with reference to the international scientific literature and guidelines by two experienced researchers and malnutrition experts (SB, DE). During this process, the individual items were evaluated to assess their applicability in nursing homes, hospitals and in home care based on experiences from practice in these fields.

The ESPEN guideline on clinical nutrition and hydration in geriatrics as well as the ESPEN guideline on definitions and terminology of clinical nutrition were mainly used as a basis for the updating and adaptation process [6,8]. The original English version of the KoM-G contained 19 items; of these, 3 items remained the same, and 9 items were changed. Additionally, 7 items were deleted, and 6 new items were generated. All items of the original KoM-G are multiple-choice items with 6 answer possibilities (including “I don’t know”). The number of answer options per item was reduced from six (6) to five (5) in order to make the questionnaire easier to understand. Furthermore, various other studies reporting on the development of knowledge instruments had 3–4 answer options (including “I don’t know”) [26,27,28]. All changes were documented and are shown in Table 1. The result was the KoM-G 2.0 questionnaire consisting of 18 items.

After the update and adaptation, a Delphi study was performed to evaluate the content validity of the English KoM-G 2.0. We included an international heterogeneous sample of experts with experience in the field of nutrition and geriatrics from acute care, nursing homes and home care. A total of 27 experts from four European countries were purposively sampled and invited to take part in the study. The 16 experts who took part (i.e., response rate of 59.3%) were dietitians, medical doctors, nutrition scientists and nursing staff from The Netherlands, Germany and Austria, enabling us to cover all three intended settings. Since the Delphi study was performed online and no identifying details were collected, no further details about the experts are available.

The Delphi study was performed in the form of an online survey, and a link to this survey was sent to the participants. The participants were asked to judge the relevance, understandability and appropriateness of each item and the answers on a four-point Likert-type scale (1 = not relevant; 2 = to some extent relevant; 3 = rather relevant; 4 = very relevant). Furthermore, they had to indicate if the items should be included in the final questionnaire or not. The Item-Content Validity Index (I-CVI) as well as the Scale-Content Validity Index/Average (S-CVI/Ave) were calculated based on the ratings of relevance. The I-CVI was expected to exceed 78%, and the S-CVI/Ave was expected to exceed 90% [29]. In the first Delphi round, the I-CVI ranged between 75.1% and 100%. After deleting two items with an I-CVI below 78%, the final KoM-G 2.0 consisted of 16 items with I-CVIs between 81.3% and 100% and a S-CVI/Ave of 94.5% (Table 2). Since the results (I-CVI and S-CVI/Ave) corresponded to the recommended values [29], no second Delphi round needed to be performed.

#### 2.1.2. Part 2: Cross-Cultural Adaptation of the KoM-G 2.0 to the German, Czech, Dutch and Turkish Languages

In the second part of the study, the 16 items included in the English KoM-G 2.0 were cross-culturally adapted and translated into the German (AT), Czech (CZ), Dutch (NL) and Turkish (TR) languages. The processes of cross-cultural adaptation and translation were carried out in accordance with the guidelines and recommendations of Beaton et al. [21], Epstein et al. [20] and Sousa and Rojjanasrirat [30]. The same steps were followed for the adaptation into all four languages, and regular meetings with all country project leaders were held. If questions arose or ambiguities were identified, the country project leaders consulted the Austrian project leader.

The process of cross-cultural adaptation included the following four steps:Forward translation: The KoM-G 2.0 was forward translated from the English language into the target languages (AT, CZ, NL, TR) by two independent persons. The mother tongue of these persons was the target language. One of the translators per country was familiar with the topic and one was not. In the subsequent consensus meetings, which were organized by the project leader for each country, the two translated versions of the questionnaire were checked for ambiguities, discrepancies in word usage, sentence construction and meanings. These meetings were held to reach a consensus (V1) between the two translators and the country project leader. Intensive discussion of all translated items and differences were held. The meetings took about 2–3 h and resulted in a final forward translation (V1) of the questionnaire. The changes and decisions were documented in a file.Back translation: The accepted consensus version (V1) was then back translated by two additional independent translators into the English language. The mother tongue of the translators was English, and one was familiar with the topic, and one was not. The results were two back translations (V2, V3) of the questionnaire.Expert committee: A national expert committee met in each country and combined the three versions (V1, V2, V3) in an online meeting. All four translators, the country project leader and the Austrian project leader took part. The translation of each item in the KoM-G 2.0 was discussed. These meetings took between 1.5 and 3 h. Finally, a consensus was reached and a pre-final version (V4) of the questionnaire was developed. Again, all decisions were documented.Pilot test: A pilot test was subsequently performed with nursing staff in hospitals, nursing homes and home care organisations in each country. The nursing staff was purposively sampled. According to Beaton et al. [21] our aim was to reach 30–40 participants per country. The pilot test was performed between July 2020 (AT) and January 2021 (TR) with 10 (NL) to 42 (CZ) participants. The participants answered the multiple-choice items of the translated KoM-G 2.0 questionnaires and rated the understandability of the instructions and items using a dichotomous scale (clear and unclear) via an online survey. Items that could not be understood by at least 20% of the participants had to be re-evaluated. Between 96.9% (NL) and 97.8% (AT) of participants rated the items as understandable; therefore, there was no need to change the items based on the outcome of the pilot study (Table 3). Nevertheless, two items were slightly changed in the CZ questionnaire in terms of the wording to increase the understandability for nursing staff [20,21,30].

## 3. Discussion

The content validity of an instrument is regarded as the most important psychometric property, because it is extremely important that instrument items are both relevant and comprehensive with respect to the construct of interest and the target population [31]. In the current study, we used the Delphi technique with a panel of 16 international experts in malnutrition with different professional backgrounds, asking them to evaluate the content validity. The S-CVI/Ave of 94.5% obtained is regarded as quite high [29]. These findings lead us to conclude that the items are highly relevant with respect to malnutrition knowledge among nursing staff. A comparison of the content validity of the original KoM-G with that of the KoM-G 2.0 found that content validity was improved. This is due to the fact that the number of experts included in the Delphi study increased, leading to more convincing results, and that the S-CVI/Ave of the updated questionnaire is higher than that of the original questionnaire (91.0% vs. 94.5%) [22].

Many different recommendations for the cross-cultural adaptation of questionnaires appear in the international literature, but the lack of evidence limits the potential to give clear recommendations [20,32]. For this reason, we decided to use various recommendations as a basis in this study. Nevertheless, we followed a structured process during the cross-cultural adaptation process [20], which resulted in appropriate instruments that can be widely used. The outcome of the cross-cultural adaptation is regarded as a strength of this study, because instruments have often been translated for use in other languages but cultural adaptation has not been performed [21]. Solely translating the instrument is not sufficient, since an instrument needs to be adapted culturally to maintain the content validity of the instrument at a conceptual level across different cultures.

The pilot study is regarded as a crucial step in the cross-cultural adaptation process [20,21]. In our study, we performed a pilot test in each country, and the test results reveal that the KoM-G 2.0 is highly understandable. Compared to the KoM-G, the KoM-G 2.0 is shorter and thus more consolidated, and therefore probably even better understandable due to the lower number of items (n = 16) and the reduction from six to five answer options per item. The high understandability of the KoM-G 2.0 is regarded as very important, since this is a precondition for the subsequent psychometric evaluation.

To our knowledge, the KoM-G 2.0 is the most up-to-date questionnaire to assess malnutrition knowledge among nursing staff in various settings. All parts of this current study were conducted by following a structured and standardized approach. The questionnaire was updated, cross-culturally adapted, and translated into four languages and the content validity was evaluated. This is a major strength of this study. The last and remaining step in the cross-cultural adaptation process is the comprehensive psychometric evaluation the KoM-G 2.0 [20,21,30]. Therefore, a factor analysis as well as the analyses of item validity (item difficulty, item discrimination, differential item functioning) and internal consistency are currently in process and will be published separately [31]. We are aware that the psychometric evaluation of instruments is a complex undertaking which requires a considerable number of different analyses and is therefore not completed with only one study [21,31,33]. Therefore, we also intend to conduct further analyses of the psychometric properties, like equivalence or cross-cultural validity, within a separate study. If psychometric properties are found to be adequate, the KoM-G 2.0 can be used to assess malnutrition knowledge among nursing staff in various settings and countries. It would also be of interest to perform pilot tests of its practicability and to test the psychometric properties of the KoM-G 2.0 with nursing students, to determine if adequate knowledge transfer on the topic of malnutrition in older adults is currently already included in the basic education of nurses.

The interest in the KoM-G and the KoM-G 2.0 is high among members of the scientific community. For instance, the KoM-G has already been translated and cross-culturally adapted to the Italian language [34]. Therefore, we advise the cross-cultural adaptation and translation of the KoM-G 2.0 for use in other countries, which would allow comparisons to be made of nutritional knowledge among more countries and would enable us to learn from each other. Nevertheless, this can only be carried out if the psychometric properties of the adapted KoM-G are sufficient. The results for different cultures would support the modification of education programs or the development of new tailored education programs to improve knowledge. Furthermore, the KoM-G 2.0 can be used to evaluate the effectiveness of training or interventions among nursing staff. Over the long term, all these strategies will be beneficial for patients and residents.

Nevertheless, this study had certain limitations. The original KoM-G was applied in the German language but was translated into English for publication purposes. This translated English version was used as the source for the update of the original KoM-G to the KoM-G 2.0. This was necessary because the international experts in the Delphi panel required an English questionnaire and not a German one. This English source version had not been validated, which may be a possible limitation. Nevertheless, we applied a comprehensive cross-cultural translation process, which may have counteracted this limitation. Furthermore, the fact that only 10 participants took part in the pilot study in The Netherlands serves as a potential study limitation.

## 4. Conclusions

This study resulted in the development of the KoM-G 2.0, an up-to-date comprehensive questionnaire available for use in various settings and in various languages. Updating and adapting the KoM-G was both necessary and successful. Content validity is regarded as the most important psychometric property of such questionnaires, and the content validity of the KoM-G 2.0 was determined to be excellent by a large international expert panel. The KoM-G 2.0 is highly understandable and now available in the German, Czech, Dutch and Turkish languages, but needs comprehensive psychometric testing before it can be used in research and practice. This psychometric testing is currently in process and will be published separately. In the long run, research results on malnutrition knowledge among nursing staff will help health care staff in these countries and settings to learn from each other. In addition, the identified knowledge in different countries enables the development of customized training offers. Thus, it will help to improve nursing education in different European countries. Another great advantage of cross-cultural adaptation is that it helps to avoid selection bias, since the availability of cross-culturally adapted instruments reduces the disadvantage of having to exclude persons who are unable to complete a questionnaire in a certain language.

## Figures and Tables

**Table 1 nutrients-16-01374-t001:** Comparison of the KoM-G and the KoM-G 2.0.

KoM-G	KoM-G 2.0	Update Result
What are possible risk factors for malnutrition?	What are possible risk factors for malnutrition?	Remained the same
What are possible consequences of malnutrition?	What are possible consequences of malnutrition?	Remained the same
What are possible signs of malnutrition?	What are possible signs of malnutrition?	Remained the same
Why should nurses keep a food and fluid log?	In which older persons should food and fluid intake be measured?	Changed
For which residents is tube feeding appropriate?	Which statements on tube feeding and parenteral nutrition are correct?	Changed
What are signs of dehydration?	What are possible signs of dehydration?	Changed
Which indicators should be assessed in nutritional screening?	Which of the following indicators should be included in nutritional screening?	Changed
When should residents be nutritionally screened?	When should a nutritional screening be conducted in older persons?	Changed
What is a “normal” and healthy BMI (Body Mass Index) of older residents (over 65 years old)?	What is considered a low BMI in older persons?	Changed
Which professionals should be involved, when necessary, in treating malnourished residents?	Which health care professionals should be included in nutritional management?	Changed
The daily total fluid requirements of a person are…	The daily total fluid requirements of an older person…	Changed
What factors can lead to higher energy and protein requirements?	Which factors can cause increased energy and nutrient requirements?	Changed
What % of unintentional weight loss in the past 3 months is a possible sign of malnutrition?		Deleted
A residents lost 3 kg in the last month. What steps can be initiated?		Deleted
To what extent do energy and nutrients requirements change for older residents (over 65 years old)?		Deleted
Which specific nutrients requirements do residents with pressure ulcers have?		Deleted
Which factors can positively affect oral nutritional intake?		Deleted
Which factors can negatively affect oral nutritional intake?		Deleted
What interventions should be ideally done for a resident with mild dysphagia at risk of malnutrition?		Deleted
	Which statements on malnutrition risk screening are correct?	New
	Which statements on the collaboration of different health professionals in a nutritional support team are correct?	New
	What should be considered in older persons’ during food intake?	New
	Which statements about the treatment of malnutrition are correct?	New
	What is important regarding the use of oral nutritional supplements in older persons?	New
	What should be considered in nutritional therapy in older persons at the end of life?	New

**Table 2 nutrients-16-01374-t002:** The 16 items of the English KoM-G 2.0, the I-CVI and S-CVI/Ave and whether they should be included in the final questionnaire.

Number	Item	I-CVI	Inclusion in Final KoM-G 2.0?
1	What are possible risk factors for malnutrition?	100%	100%
2	What are possible consequences of malnutrition?	100%	100%
3	What are possible signs of malnutrition?	93.8%	93.9%
4	What are possible signs of dehydration?	81.3%	87.5%
5	Which statements on malnutrition risk screening are correct?	100%	100%
6	Which of the following indicators should be included in nutritional screening?	100%	100%
7	When should a nutritional screening be conducted in older persons?	100%	93.8%
8	Which statements on the collaboration of different health professionals in a nutrition support team are correct?	87.6%	93.8%
9	The daily total fluid requirement of an older person…	81.3%	81.3%
10	Which of the following factors can cause increased energy and nutrient requirements?	93.7%	87.5%
11	In which older persons should food and fluid intake be measured?	100%	87.5%
12	What should be considered during older persons’ food intake?	93.7%	93.8%
13	Which statements about the treatment of malnutrition are correct?	100%	93.8%
14	What is important regarding the use of oral nutritional supplements in older persons?	100%	93.8%
15	Which statements on tube feeding and parenteral nutrition are correct?	87.4%	93.8%
16	What should be considered regarding the nutritional therapy in older persons at the end of life?	93.7%	93.8%
**S-CVI/Ave**		94.5%	

I-CVI = Item-Content Validity Index; S-CVI/Ave = Scale-Content Validity Index/Average.

**Table 3 nutrients-16-01374-t003:** Pilot study.

	AT	CZ	NL	TR
**When was the pilot study performed?**	July–August 2020	November–December 2020	December 2020–January 2021	September–October 2020
**How many participants took part?**	n = 31	n = 42	n = 10	n = 40
**How many participants rated the items as understandable?**	97.8%	97.5%	96.9%	100%
**Were items changed and if yes, how many items?**	No changes	Two items were slightly changed in terms of wording	No changes	No changes

AT = Austria; CZ = The Czech Republic; NL = The Netherlands, TR = Turkey.

## Data Availability

The data presented in this study are available on request from the corresponding author due to privacy and ethical restrictions.

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
