# Peer review of "The Knowledge of Malnutrition—Geriatric (KoM-G) 2.0 Questionnaire for Health Care Institutions: Cross-Cultural Adaptation into German, Czech, Dutch and Turkish"

_nutrients, 2024, doi:10.3390/nu16091374_

Round 1
Reviewer 1 Report
Comments and Suggestions for Authors
The authors presented their work with the objective of creating an improved version of the Knowledge of Malnutrition-Geriatric (KoM-G) questionnaire, which included the incorporation of additional items, translation of the questionnaire to different languages, the cross-cultural adaptation, and an assessmente of content validity.
Although the paper itself presents no major issues, the main problem with this manuscript is that the authors did not conduct the necessary validation assessments that will determine whether the proposed version of the scale truly measures what it is intended to measure. At least an internal validation study would be essential, with information on internal consistency reliability, item analysis, convergent validity and test-retest reliability. From the Discussion, it appears that the authors do not have plans to conduct such validity evaluations, since they seem to be offering suggestions on further assessments of the questionnaire to whoever whishes to pick up the challenge, as might be interpreted by sentences like :"Therefore, we recommend further analysis of the psychometric properties like 230 equivalence or cross-cultural validity. It would also be of interest to perform pilot tests 231 about the practicability of the KoM-G 2.0 with nursing students". In my opinion, the publication of this unvalidated questionnaire might lead misinformed professional to replace the original KoM-G questionnaire, which has shown to possess evidence of excelent internal validity, with a version whose performance might be much poorer.
My suggestion is that the authors conduct a full internal validation study, including a confirmatory factor analysis, and preferably an external validation study as well, and publish the results if their version of the quationnaire is shown to be superior to the original one.
Reviewer 2 Report
Comments and Suggestions for Authors
Authors have updated "the knowledge of malnutrition-geriatric (KoM-G) questionaire and have culturally adapted this tool in four languages, namely German, Czeck, Dutch and Turkish languages. The methology is in accordance with standards recommendations. Papers is written well, but there are some fundamental questions to be answered:
Authors state that in part 1 of the study, they have updated this tool and adapted it for use in different settings. The main comment at this step is that, as authors acknowledge, this tool was originally developed in Austria for use in nursing staff. One may ask whether the nutritional assessments, interventions and similars tasks are carried out by nursing staff in European countries? Why dietitians/nutritionist have not been enrolled if the main aim was to use this tool in Health Care "Institutions"? Apparently authors have engaged in the same sampling error and approach bias that was exist in the original Austrian study [22], i.e, why this tool has not been developed and adapted/upgraded to be used by nutritionists as the main target group?. I appreciate that all authors are involved in nutritional studies, but that would be very nice if some experts with nutritional affiliation (or dietitians) were also involved. The second main concern is the existance of ethnic groups. For instance, only in Turkey, about 14 milion Kurdish people live. In their pilot study, it is not clear how were the participants choosen and whether these participants were representative of Turkey as a multi-ethnic country? (this is only one of many questions). It would be instrumental if authors provide these important points as a supplementary file (maybe as a table describing demographic of participants).
Minor:
Line 60, please elaborate further: only having nutritional knowledge in nursing staff, does not automatically leads to proper nutritional status in patients; many other variables (attitude, practice, budget, etc.) are involved too. It is also very important to provide any changes (in detail) in the upgraded items as a supplementary file for future studies. It is also a question why authors have stated that "data is unavailable due to privacy and ethical restrictions". That would be nice if authors state why?
Reviewer 3 Report
Comments and Suggestions for Authors
In this study, the authors revised the KoM-G questionnaire and verified its validity, as well as its validity in four languages. They also show that KoM-G2 could be used in the future to educate medical staff on nutrition. I understood the significance of the research. However, there are some places where I felt that the necessary information was missing. To help readers understand, please add information about the points below.
Line 110-114; Specifically, which items were changed and for what reason? Since this point is one of the important topics in this study, it is necessary to show it in more detail. Also, there is a statement that “The number of answers per item was reduced from six (6) to five (5) options.” Is this correct in understanding that there are options for each question item? If so, please state the options for each item.
